# GPR109A mediates the effects of hippuric acid on regulating osteoclastogenesis and bone resorption in mice

Jin-Ran Chen [1,2✉], Haijun Zhao[1,2], Umesh D. Wankhade[1,2], Sree V. Chintapalli[1,2], Can Li[3], Dongzheng Gai[3], Kartik Shankar[1,2,4], Fenghuang Zhan[3] & Oxana P. Lazarenko[1,2]

The G protein-coupled receptor 109 A (GPR109A) is robustly expressed in osteoclastic precursor macrophages. Previous studies suggested that GPR109A mediates effects of diet-derived phenolic acids such as hippuric acid (HA) and 3-(3-hydroxyphenyl) propionic acid (3-3-PPA) on promoting bone formation. However, the role of GPR109A in metabolic bone homeostasis and osteoclast differentiation has not been investigated. Using densitometric, bone histologic and molecular signaling analytic methods, we uncovered that bone mass and strength were significantly higher in tibia and spine of standard rodent diet weaned 4-week-old and 6-month-old GPR109A gene deletion (GPR109A$^{-/-}$) mice, compared to their wild type controls. Osteoclast numbers in bone and in ex vivo bone marrow cell cultures were significantly decreased in GPR109A$^{-/-}$ mice compared to wild type controls. In accordance with these data, CTX-1 in bone marrow plasma and gene expression of bone resorption markers (TNFα, TRAP, Cathepsin K) were significantly decreased in GPR109A$^{-/-}$ mice, while on the other hand, P1NP was increased in serum from both male and female GPR109A$^{-/-}$ mice compared to their respective controls. GPR109A deletion led to suppressed Wnt/β-catenin signaling in osteoclast precursors to inhibit osteoclast differentiation and activity. Indeed, HA and 3-3-PPA substantially inhibited RANKL-induced GPR109A expression and Wnt/β-catenin signaling in osteoclast precursors and osteoclast differentiation. Resultantly, HA significantly inhibited bone resorption and increased bone mass in wild type mice, but had no additional effects on bone in GPR109A$^{-/-}$ mice compared with their respective untreated control mice. These results suggest an important role for GPR109A during osteoclast differentiation and bone resorption mediating effects of HA and 3-3-PPA on inhibiting bone resorption during skeletal development.

[1] Arkansas Children's Nutrition Center, Little Rock, AR 72202, USA. [2] Department of Pediatrics, University of Arkansas for Medical Sciences, Little Rock, AR 72202, USA. [3] Myeloma Center, University of Arkansas for Medical Sciences, Little Rock, AR 72202, USA. [4]Present address: Department of Pediatrics, Section of Nutrition, University of Colorado School of Medicine, Aurora, CO 80045, USA. ✉email: chenjinran@uams.edu

The G protein-coupled receptor 109 A (GPR109A), also known as several other names such as hydroxycarboxylic acid receptor 2 (HCAR2), niacin receptor 1 (NIACR1), HM74A or PUMA-G, is expressed in a variety of cells and tissue types, and robustly in osteoclastic precursor macrophages[1,2]. This receptor may sense gut and liver metabolites, modulating cell signaling that is coupled to energy and lipid metabolism as well as immune cell function[3]. GPR109A is now appreciated as an important target of niacin (also known as vitamin B3 and nicotinic acid), which led to wide investigations on both GPR109A and niacin on their clinical values for the treatment of dyslipidemia and to increase HDL cholesterol[4,5]. It has been suggested that full agonists of GPR109A include D-β-hydroxybutyric acid and β-hydroxybutyrate[6,7], butyric acid and butyrate[8], and niacin[9]. However, whether these agonists exert similar clinical properties and tissue-specific roles are not clear. Although niacin is known as a lipid-lowering drug and has been in clinical use for more than 40 years[10], none of these compounds have been documented as to beneficial or negative effects on bone development, metabolism or remodeling.

In mice lacking GPR109A, the nicotinic acid-induced decreases in free fatty acid (FFA) and triglyceride plasma levels were abrogated[1]. It has also been shown that the anti-lipolytic effect of nicotinic acid involves the inhibition of cyclic adenosine monophosphate (cAMP) accumulation in adipose tissue through a G (i)-protein-mediated inhibition of adenylyl cyclase[11]. From several other studies, using GPR109A deletion cell and mouse models, roles for GPR109A on regulating immune cell, hepatocyte and gut epithelial cell function were proposed[12]. A recent report showed that GPR109A gene deletion resulted in increased liver size and body weight[13].

We have proposed that GPR109A also plays a role in bone cell function and bone homeostasis. For instance, GPR109A is expressed in murine osteoclasts (http://biogps.org/#goto=genereport&id=338442) and can be induced significantly by inflammatory insults in murine osteoclast precursor bone marrow macrophages. From our previous studies using cell models in which GPR109A expression was modulated[14,15], we have provided evidence that GPR109A plays a role on bone forming cell differentiation. However, until now, the role of GPR109A on osteoclast activity, differentiation and bone resorption in vivo has not been determined.

Multinucleated osteoclasts are essentially macrophages in bone, they are derived from hematopoietic lineage[16]. Suppressing osteoclastogenesis or mature osteoclast activity are proposed effective therapeutic approaches for bone-destructive diseases, such as osteoporosis and rheumatoid arthritis. However, osteoclastic bone resorption is also an important physiologic function to shape the bone during early development and adulthood. It is well-known that osteoclast differentiation and its activity are regulated by various molecular signals, such as RANKL (receptor activator for nuclear factor κB ligand), a well investigated essential factor involved in both processes[17]. Since other cell lineages derived from hematopoietic precursors share similar signaling pathways, it is not yet clear how specific intervention is to osteoclast formation; therefore, identifying novel molecules that can regulate osteoclastogenesis has been an important clinical goal.

In the current report, we set to determine the impact of GPR109A on bone development and bone cell function using a GPR109A gene deletion mouse model. In addition, we hypothesized that natural plant-derived phenolics such as hippuric acid (HA) and 3-(3-hydroxyphenyl) propionic acid (3-3-PPA) act as inhibitors of GPR109A to suppress osteoclast differentiation and bone resorption. These phenolic acids are gut microflora-derived metabolites of polyphenols found in the circulation following consumption of blueberry (BB) and other fruits, vegetables and coffee[18]. We have previously provided evidence that the actions of HA and 3-3-PPA on bone cells is mediated in part through mechanisms involving binding to GPR109A in cell membranes[19,20], and they were able to stimulate osteoblast differentiation and activity. Thus, in the current study we tested if whole-body loss of GPR109A modifies the actions of HA or BB treatment on bone outcomes.

## Results

**Increased bone volume and trabecular number in GPR109A⁻/⁻ mice.** BioGPS shows that GPR109A is expressed in a variety of tissues, with the highest expression being in bone marrow macrophages (BioGPS - your Gene Portal System)[21]. We isolated the RNA from a variety of tissues taken from a 4-week-old wild type mouse and RNA from two mouse-origin cell lines. Using standard PCR and real-time PCR, GPR109A mRNA expression levels were confirmed in those tissues and cell lines. In vertebrae, abdominal fat (Ab. Fat), small intestine (Sm. Intest.) and RAW246.7 cells, GPR109A mRNA was highly expressed (Fig. 1a). It was also expressed in tibia, bone marrow cells (BMC), brain, heart, kidney, spleen and mouse-origin stromal cell line ST2 cells, but less expressed in liver and gastric muscles (G. Mus.)[22,23] (Fig. 1a).

At 4 weeks of age, micro-CT on tibia showed significantly higher actual bone volume (BV) and bone mineral density (BMD), trabecular number (Tb.N) and bone volume per total tissue volume (BV/TV) in the knockout mice compared with wild type mice (Fig. 1b). However, trabecular bone separation (Tb.Sp) was significantly lower in GPR109A gene deletion mice (Fig. 1b) indicating decreased ongoing bone resorption. As shown in Fig. 1c of representative micro-CT images (three animals per group), we did not observe significant changes on cortical micro-CT parameters; this may due to the young age (four weeks old) of the mice that were analyzed. At this age, no differences were found between GPR109A gene deletion (GPR109A⁻/⁻) and wild type mice on other basic growth parameters such as body weights and long bone length. Increased bone mass phenotype of GPR109A⁻/⁻ mice was also found in vertebrae of male and female mice (Fig. 2) using micro-CT analysis. As shown in Table 1, BV/TV and Tb.N were significantly increased in GPR109A⁻/⁻ mice, but Tb.Sp was significantly decreased compared with their respective male and female wild type mice. In males, degree of anisotropy (DA) was significantly higher in GPR109A⁻/⁻ mice, and in females, BS/TV (bone surface density) was significantly higher in GPR109A⁻/⁻ mice compared with their respective control wild type mice.

**Decreased osteoclastogenesis in GPR109A⁻/⁻ mice.** Using micro-CT on femurs, bone strength was compared between wild type and GPR109A⁻/⁻ male and female young (4-week-old) and adult (6-month-old) mice using three-point bending test after positioning. In 4-week-old mice, moment of inertia and outer cortical thickness, but not anterior/posterior depth were significantly higher in male GPR109A⁻/⁻ mice compared with their wild type (Fig. 3a). We did not observe differences of those parameters in females between wild type and GPR109A⁻/⁻ mice (Fig. 3a). Maximum loading force curves were steeper in samples from GPR109A⁻/⁻ males and females compared with their respective gender wild type mice (Fig. 3a). Regardless of gender, bending test parameters of yield load, stiffness, and modulus of elasticity were significantly higher in GPR109A⁻/⁻ mice compared with wild type mice (Fig. 3c). In 6-month-old mice, moment of inertia, anterior/posterior depth, and outer cortical thickness were found significantly higher in female GPR109A⁻/⁻ mice compared

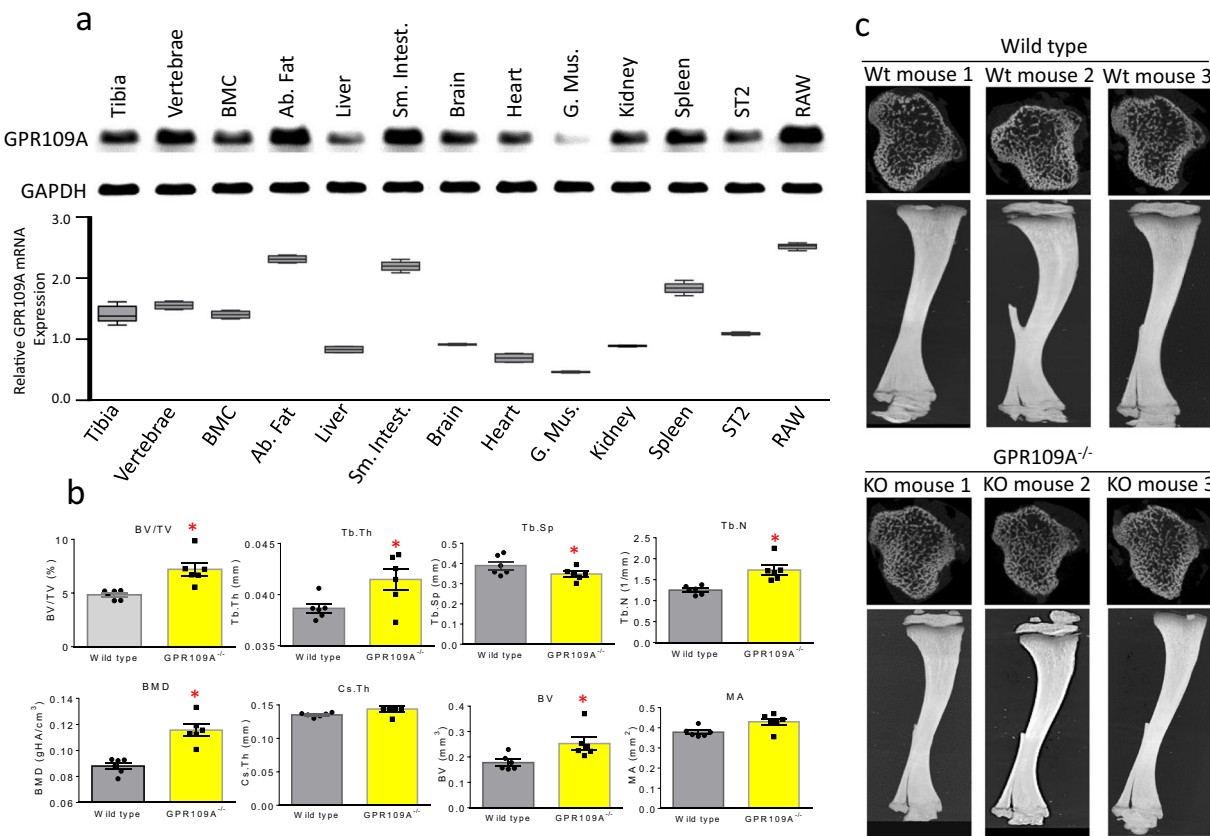

**Fig. 1 Increased bone mass phenotype in GPR109A$^{-/-}$ mice. a** PCR for GPR109A mRNA expression in tissues taken from a 4-week-old male mice: tibia, vertebrae, bone marrow cells (BMC), abdominal fat (Ab. Fat), liver, small intestine (Sm. Intest.), brain, heart, gastric muscles (G. Mus.), kidney, spleen and mouse-origin stromal cell line ST2 cells and macrophage cell line RAW246.7 cells (RAW), and control GAPDH mRNA expression. Box and whiskers graph under gel is real-time PCR results for relative GPR109A mRNA expression in those tissues and cells. **b** Micro-CT measured parameters from 4-week-old male wild type and GPR109A$^{-/-}$ mouse groups. BV/TV, bone volume/total tissue volume; Tb.Th, trabecular thickness; Tb.Sp, trabecular separation; Tb.N, trabecular number; BMD, actual bone mineral density; Cs.Th, cortical thickness; BV, actual bone volume; MA, medullary area; BA/TA, bone area/total area. Data are expressed as mean ± SD ($n$ = 6 per group). *$p$ < 0.05 by $t$-test. **c** Representative micro-CT images of the proximal tibia from three samples from each group of mice.

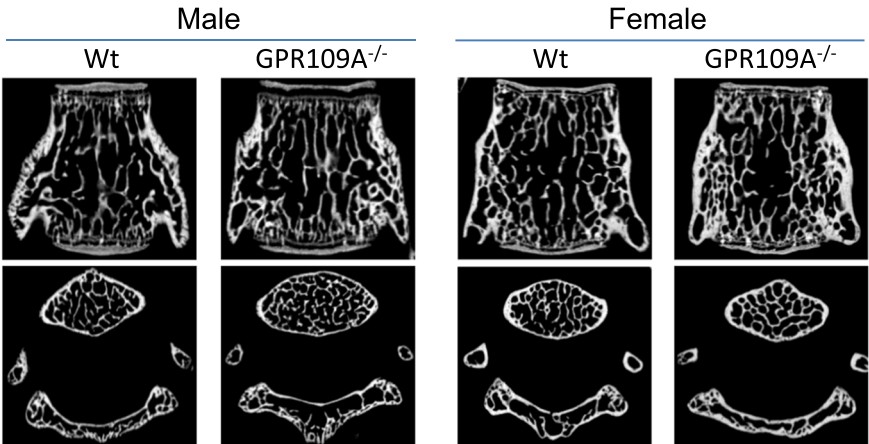

**Fig. 2 Micro-CT on L5 vertebrae from male and female GPR109A$^{-/-}$ and wild type mice.** Representative micro-CT images (sagittal view above, axial view below) of the L5 vertebrae from one sample from 4-weeks-old male and female of each group of GPR109A$^{-/-}$ and wild type (Wt) mice, white color shows trabecular bone.

with their wild type (Fig. 3d), while outer cortical thickness was found significantly higher in male GPR109A$^{-/-}$ mice compared with their wild type (Fig. 3d). While loading force curves were steeper in samples from GPR109A$^{-/-}$ males and females compared with their respective gender wild type mice (Fig. 3e), bending test parameters of yield load, stiffness, and modulus of elasticity were only significantly higher in GPR109A$^{-/-}$ female mice compared with their wild type mice (Fig. 3f). To confirm increased bone mass phenotype of GPR109A$^{-/-}$ mice, we next performed peripheral quantitative CT scan (pQCT) on tibia. As shown in the

**Table 1 Micro-CT parameters on vertebra of wild type and GPR109A-/- male and female mice.**

| | Male | | | Female | | |
|---|---|---|---|---|---|---|
| | Wt | GPR109A$^{-/-}$ | p | Wt | GPR109A$^{-/-}$ | p |
| Trabecular | | | | | | |
| BV/TV (%) | 16.8 ± 1.66 | 19.7 ± 1.98* | 0.02 | 19.05 ± 2.13 | 21.95 ± 1.55* | 0.02 |
| BS/BV (1/mm) | 102.0 ± 6.4 | 104.4 ± 2.7 | 0.23 | 90.1 ± 7.98 | 86.9 ± 6.1 | 0.25 |
| BS/TV (1/mm) | 18.4 ± 2.7 | 20.6 ± 1.7 | 0.08 | 17.0 ± 1.2 | 21.9 ± 1.5* | 0.02 |
| Tb.N (1/mm) | 4.88 ± 0.48 | 5.78 ± 0.52* | 0.01 | 4.80 ± 0.32 | 5.67 ± 0.50* | 0.01 |
| Tb.Th (mm) | 0.036 ± 0.002 | 0.034 ± 0.001 | 0.08 | 0.04 ± 0.002 | 0.041 ± 0.003 | 0.20 |
| Tb.Sp (mm) | 0.14 ± 0.02 | 0.12 ± 0.009* | 0.03 | 0.16 ± 0.01 | 0.14 ± 0.006* | 0.03 |
| DA | 0.44 ± 0.06 | 0.53 ± 0.04* | 0.015 | 0.45 ± 0.02 | 0.42 ± 0.05 | 0.06 |
| TV (mm$^3$) | 0.30 ± 0.02 | 0.32 ± 0.01 | 0.10 | 0.35 ± 0.045 | 0.35 ± 0.08 | 0.36 |
| BV (mm$^3$) | 0.056 ± 0.02 | 0.063 ± 0.007 | 0.19 | 0.068 ± 0.013 | 0.082 ± 0.02 | 0.10 |

Micro-computed tomography (CT) measurements of trabecular of vertebra bone L5 from 4-weeks-old male and female mice were evaluated using a Skyscan microCT scanner (SkyScan 1272, Bruker. com) at 8 μm pixel size with an X-ray source power of 60 kV and 166 μA and integration time of 950 ms. BV/TV, Percent bone volume; BS/BV, Bone surface / volume ratio; BS/TV, Bone surface density; Tb.N, Trabecular number; Tb.Th, Trabecular thickness; Tb.Sp, Trabecular separation; DA, Degree of anisotropy; TV, Tissue volume; BV, Bone volume. *$p < 0.05$ versus Wt versus GPR109A$^{-/-}$, $n = 6$, mean ± SD.

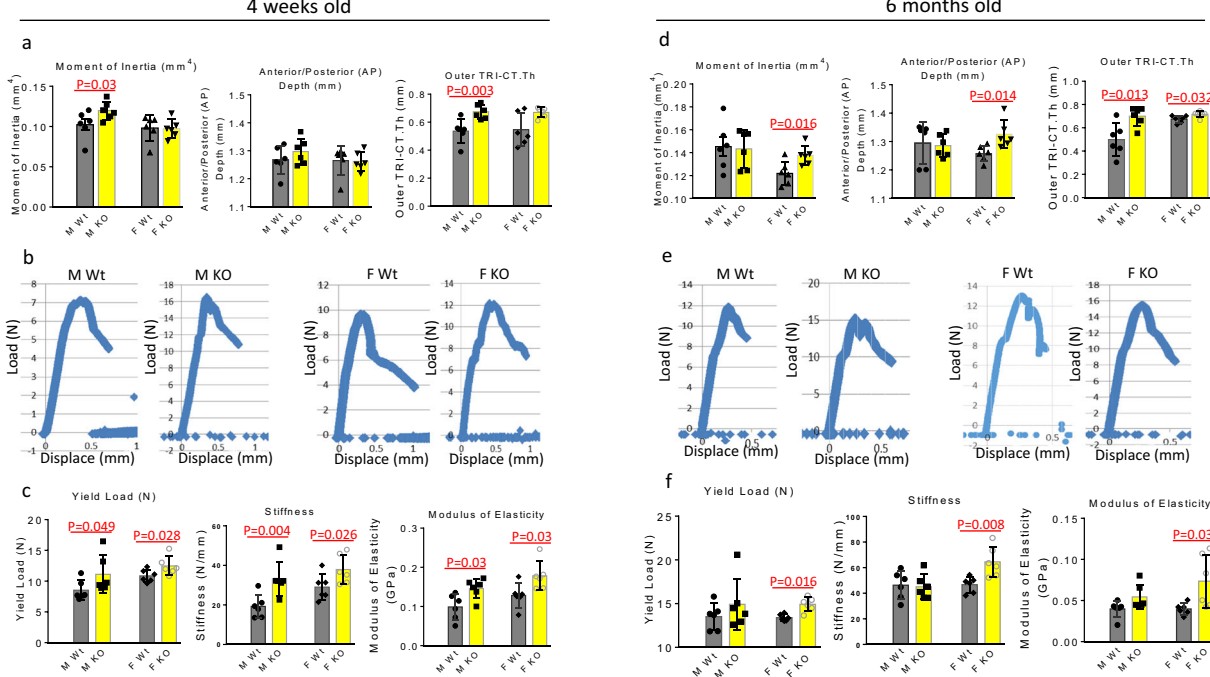

**Fig. 3 Biomechanical testing of mouse femurs using three-point bending. a** Three different Micro-CT parameters of femurs from 4 weeks old and, **d** 6 months old mouse before three-point bending tests. **b** Represented association curves between inverse load (y axis) and inverse displace (X axis) from three-point bending tests of femurs from 4 weeks old and, **e** 6 months old mice. **c** Bone strength parameters after three-point bending tests of 4 weeks old and, **f** 6 months old mouse femurs. Male wild type (M Wt), male GPR109A$^{-/-}$ (M KO), female wild type (F Wt), female GPR109A$^{-/-}$ (F KO) mice. Red p = numbers mean significantly different versus respective wild type mice by t-test, $n = 6$, mean ± SD.

representative pQCT images in Fig. 4a (three images per group are shown), color changes from black to white indicate changes in trabecular bone density from lower to higher; clearly, bone density in GPR109A$^{-/-}$ mice was higher than control wild type mice. In accordance with color changes on pQCT images, pQCT parameters of trabecular density increased about 25% in GPR109A$^{-/-}$ mice compared to their wild type controls (Table 2), body weight and tibia length were not significantly different. While total BMD significantly increased in GPR109A$^{-/-}$ mice, cortical BMD did not show significant changes in this age of GPR109A$^{-/-}$ mice compared with their control wild type mice (Table 2). SSI polar, a calculated parameter of pQCT analysis reflecting bone bending strength, was significantly increased in GPR109A$^{-/-}$ mice

compared to their wild type controls (Table 2), indicating bone quality is better in GPR109A$^{-/-}$ mice. Right tibia was cryosectioned for bone histological study. TRAPase staining was performed on cryosectioned tibia (three samples per group are presented in Fig. 4b), and pink stained TRAPase-positive osteoclastic cells in samples from GPR109A$^{-/-}$ mice were clearly fewer than samples from wild type controls (Fig. 4b). Osteoclast numbers were significantly decreased (Fig. 4c), while osteoblast numbers (Fig. 4d) were significantly increased in GPR109A$^{-/-}$ mice compared with their wild type controls. In accordance with this, bone marrow plasma osteoclastic resorption marker CTX-1 showed significantly lower levels in mice from GPR109A$^{-/-}$ compared with their wild type controls (Fig. 4e), on the other hand, serum bone formation

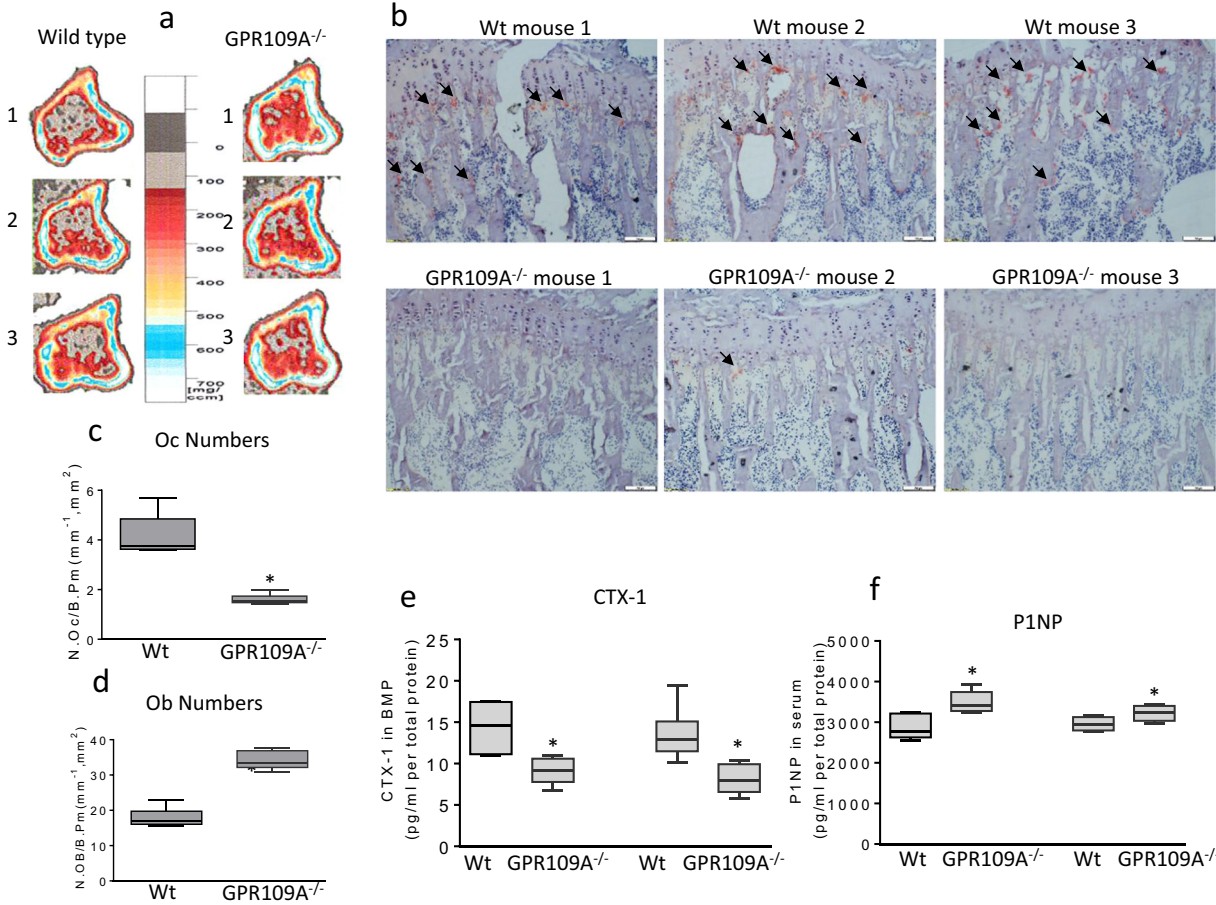

**Fig. 4 Increased bone mass in GPR109A$^{-/-}$ mice is associated with decreased osteoclast numbers. a** Representative sagittal views of quantitative pQCT analysis of one slice of the proximal tibial from three male samples from each group of mice. Bar in the middle shows color changes from black to white indicating lower to higher bone density. **b** Bone histology on cryosectioned tibia close to growth plate for TRAPase staining, images of magnification 40x from epifluorescent microscope (model BH-2, Olympus) showing three representative male samples from each group. Black arrows show pink TRAPase positively stained osteoclastic cells on bone surface. **c** Osteoclast and **d** osteoblast numbers counted per bone from wild type (Wt) and GPR109A$^{-/-}$ mice. **e** Bone marrow plasma resorption marker CTX-1 and, **f** serum P1NP levels by ELISA. Left two bars are from male wild type and GPR109A$^{-/-}$ mice and right two bars from age-matched (4-weeks-old) female wild type and GPR109A$^{-/-}$ mice, *$p < 0.05$ versus respective wild type mice by $t$-test, $n = 6$, mean ± SD.

**Table 2 Tibia pQCT parameters.**

|  | Wild type | GPR109A$^{-/-}$ | P value |
|---|---|---|---|
| Body weight (g) | 16.2 ± 1.66 | 16.9 ± 1.20 | 0.32 |
| Tibia length (mm) | 15.8 ± 0.65 | 15.6 ± 0.90 | 0.46 |
| Total density (mg/ccm) | 333.9 ± 30.4 | 367.8 ± 37.1 | 0.019 |
| Trabecular density (mg/ccm) | 143.1 ± 21.4 | 195.8 ± 27.9 | 8.3E-05 |
| Cortical density (mg/ccm) | 489.4 ± 46.5 | 508.1 ± 47.5 | 0.19 |
| SSI polar (mm$^3$) | 0.22 ± 0.04 | 0.32 ± 0.09 | 0.003 |

Boday weights and tibial length and Pqct parameters in 4-weeks-old wild type and GPR109A-/- (knock out) mice. Data presented as mean ± SD, p value was detected by t-test. SSI polar, strength-strain index polar.

marker P1NP showed significantly higher levels in mice from GPR109A$^{-/-}$ compared with their wild type controls (Fig. 4f).

We next isolated bone marrow cells and suspended them in culture medium for 48 h. Non-adherent bone marrow cells considered as osteoclast precursors were then re-cultured in the presence of 50 ng/ml of RANKL. Photomicrographs of three representative samples from each group displaying osteoclast morphology after TRAPase staining (Fig. 5a). Osteoclast number per well, done in triplicate for each mouse, was significantly

decreased in hematopoietic non-adherent bone marrow cells from GPR109A$^{-/-}$ mice (Fig. 5b). Cells were re-cultured for evaluation of osteoclast resorptive activity in Corning osteo-assay plates. Figure 5c presents resorption pits in white from three representative samples from each group. Figure 5d, depicting percentage of bone resorption area per well done in triplicate for each mouse, showed significantly lower osteoclast resorptive activity from GPR109A$^{-/-}$ mice. Total RNA was isolated from hematopoietic non-adherent bone marrow cells; real-time PCR showed significantly lower expression levels of mRNA of osteoclast markers, including NFκB, TRAP, Cathepsin K, DNMT3a and Ezh2 in samples from GPR109A$^{-/-}$ mice (Fig. 5e). Decreased DNMT3a and Ezh2 mRNA expression in samples from GPR109A$^{-/-}$ mice suggested that GPR109A might be involved in an epigenetic regulation pathway during osteoclast differentiation.

**Gut microbiota β-diversity and differences in family level between GPR109$^{-/-}$ and wild type mice.** The above results suggested that GPR109A plays a direct role on osteoclastogenesis; however, systemic GPR109A gene deletion may also indirectly effect bone tissue development. Previously, changes to the gut microbiome have been linked to shifts in bone development and

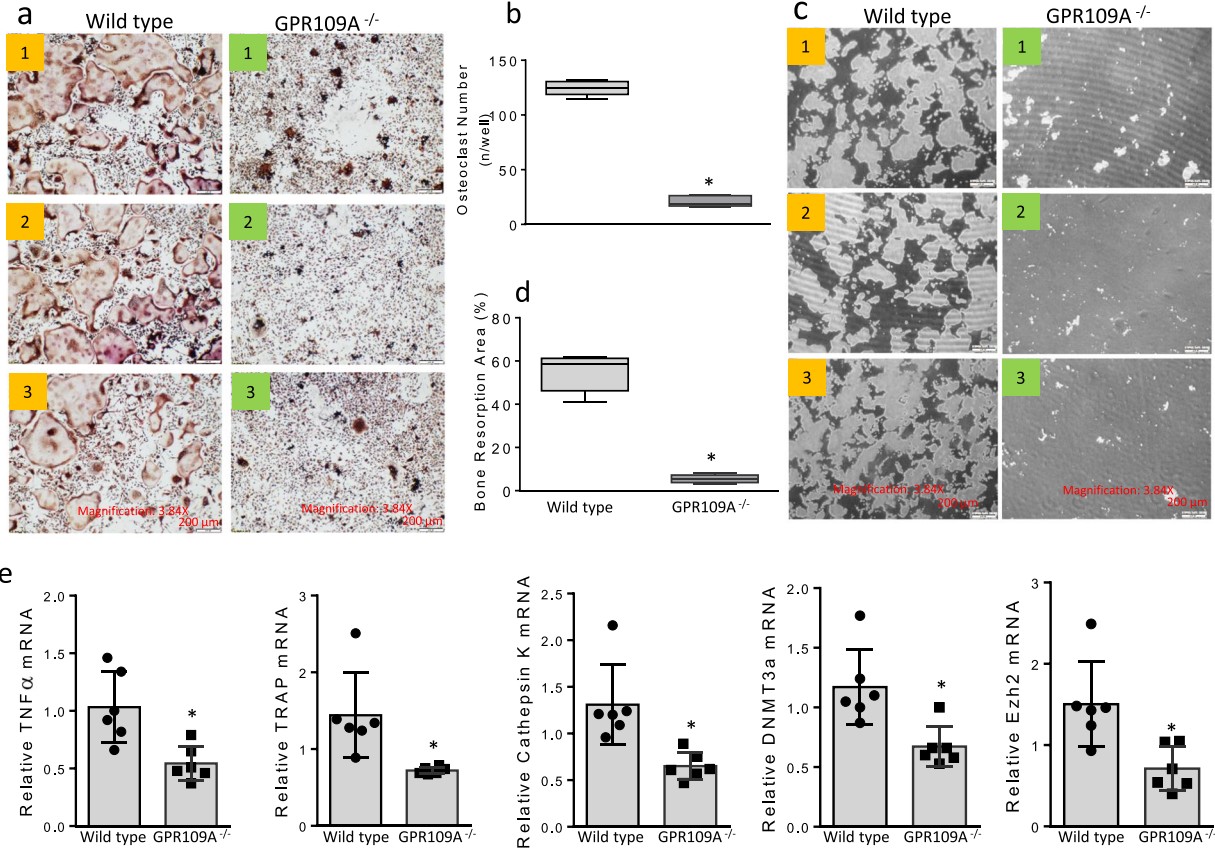

**Fig. 5 Decreased osteoclastogenesis in GPR109A$^{-/-}$ mice in ex vivo non adherent bone marrow cell culture. a** Bone marrow cells were isolated from 4-week-old male wild type and GPR109A$^{-/-}$ mice and suspended for 48 h, non-adherent bone marrow cells were re-cultured in the presence of 50 ng/ml RANKL for 5 days. Representative pictures showing osteoclast morphology from 3 different individual wild type or GPR109A$^{-/-}$ mice after TRAPase staining. **b** Osteoclast number per well with triplicates for samples from each mouse. **c** Non-adherent bone marrow cells from wild type and GPR109A$^{-/-}$ mice were cultured for osteoclast resorptive activity in corning osteo assay plates. Pictures showing resorption pits (white spots/areas) without Von Kossa staining. **d** Percentage of bone resorption area per well with triplicates for samples from each mouse. **e** Total RNA was isolated from non-adherent bone marrow cells from wild type and GPR109A$^{-/-}$ mice, real time RT-PCR shows relative NF$k$B, TRAP, Cathepsin K, DNMT3a, Ezh2 mRNA expression. Data are shown as the means ± SD of $n = 6$, gene expression was relative to housekeeping gene GAPDH. *$p < 0.05$ versus wild type control by $t$-test.

osteoclast-osteoblast functions, through signals that are not fully known[24]. Some genes encoded proteins were able to shape the microbiome by regulating the availability of nutrients, therefore to be linked to the processes of development of metabolic diseases of the host[25,26]. We tested if loss of GPR109A impacts microbiome ecology leading to effects on bone development. We analyzed the gut microbiota composition by 16 S rRNA amplicon sequencing of cecal contents and observed distinct differences in the microbial taxa associated with GPR109A gene deletion in 4 wk old male and female mice. For this analysis, we were also interested in understanding the microbial ecology. Using isolated DNA from cecal contents and illumina platform, we sequenced the V4 region of bacterial DNA. Sequenced reads were analyzed by using a standardized pipeline which yielded several outcomes including alpha and beta diversity and differential abundance at multiple taxonomic levels. β-diversity (a global measure of microbiome composition) showed a clear separation based on genotype using both distance and dissimilarity based metrics (Supplemental Figure 1a). Several phyla were affected with the absence of GPR109A. Two main phyla, *Bacteroidetes* and *Firmicutes*, were significantly altered because of the absence of GPR109A (Supplemental Figure 1b). *Firmicutes* to *Bacteroidetes* ratio was significantly lower in GPR109A$^{-/-}$ mice. In addition, large changes were observed in the abundance of other phyla (*Proteobacteria* and *Actinobacteria*) in GPR109A$^{-/-}$ mice

compared to wild type animals (Supplemental Figure 1b). Consistent with the results in male GPR109A$^{-/-}$ mice, changes of gut microbiota composition in GPR109A$^{-/-}$ mice were significantly different than those in female wild type controls (Supplemental Figure 1c,d).

**Signaling cascades involved inhibitory effects of HA and 3-3-PPA on osteoclastogenesis through suppression of GPR109A expression.** We have recently reported an in vitro study suggested that HA and 3-3-PPA inhibit osteoclastogenesis and bone resorption through suppressing GPR109A in pre-osteoclasts[15]. To further search possible membrane GPR109A-mediated actions of HA and 3-3-PPA on inhibition of osteoclastogenesis, we performed an in ex vivo study. Non-adherent bone marrow hematopoietic cells were isolated from four 4-week-old male mice. Cells were cultured in 6-well plates and were treated with or without HA or 3-3-PPA in the presence or absence of RANKL for 2 days. Proteins were isolated for Western blot analysis. It was very clear that RANKL increased NF$k$B, NFATc1, MMP9, and Cathepsin K protein expression (Fig. 6a). Compared to untreated controls, it was not clear if HA and 3-3-PPA inhibited NF$k$B and Cathepsin K protein expression (Fig. 6a). However, HA and 3-3-PPA inhibited RANKL-induced protein over-expression of NF$k$B, NFATc1, MMP9, and Cathepsin K (Fig. 6a). It was reported that

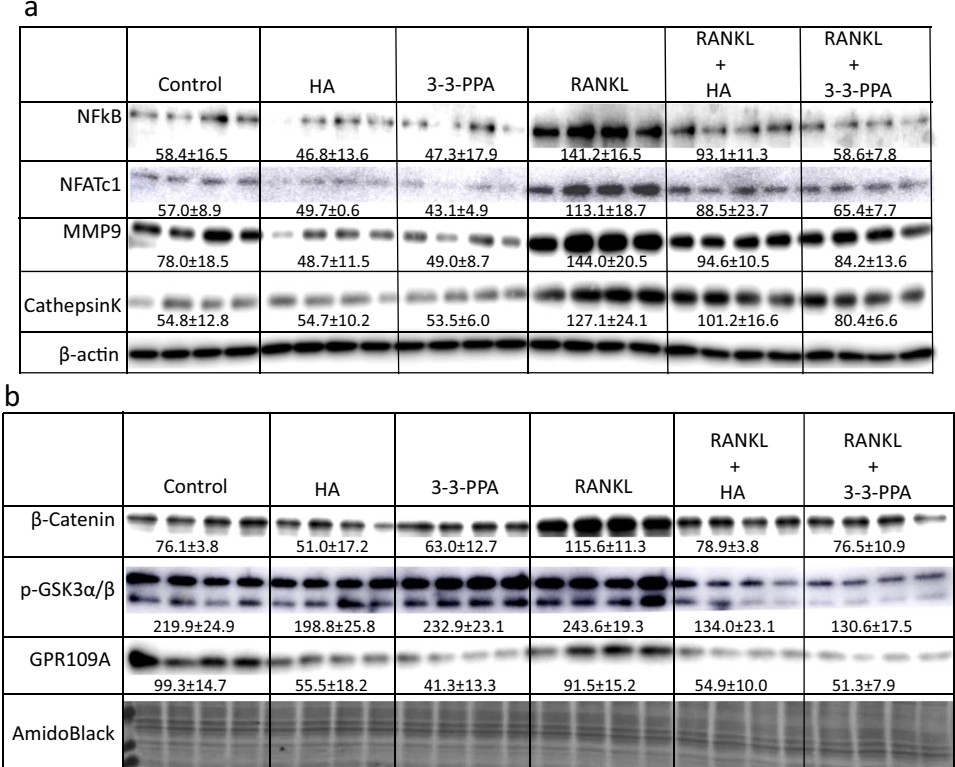

**Fig. 6 GPR109A-mediated inhibition of RANKL-induced osteoclastogenic signaling by HA and 3-3-PPA.** Non-adherent bone marrow cells were isolated from 4-week old male wild type mice, cells were treated with HA or 3-3-PPA (HA, 60 µg/dL; 3-3-PPA, 100 µg/dL) in the presence or absence of 50 ng/ml RANKL for 48 h. **a** Western blot shows RANKL activated NFκB, NFATc1, MMP9 and Cathepsin K protein expression. Both HA and 3-3-PPA inhibited RANKL-induced protein expression. **b** Western blot shows inhibition of β-catenin and GSK3α/β and GPR109A expression by HA and 3-3-PPA in the presence of 50 ng/ml RANKL. Numbers under blots are mean ± SD band intensities relative to loading controls.

Wnt/β-catenin is a signaling molecule that is involved in regulation of both osteoclast and osteoblast activity and differentiation[27]. We therefore checked if HA and 3-3-PPA regulate β-catenin signaling in non-adherent bone marrow hematopoietic pre-osteoclasts. We found both HA and 3-3-PPA blocked RANKL-induced β-catenin protein over-expression, and phosphorylation of GSK3β (Fig. 6b). Notably, HA and 3-3-PPA significantly inhibited GPR109A protein expression in those non-adherent bone marrow hematopoietic pre-osteoclastic cells (Fig. 6b). We did not see induction of GPR109A expression by RANKL, but it was obvious that with the combination of RANKL and HA or 3-3-PPA, GPR109A protein expression was below control levels (Fig. 6b).

**BB diet and HA promote bone development by inhibition of osteoclast bone resorption but had no additional effects in GPR109A$^{-/-}$ mice.** Considering the apparent interactions of diet-derived phenolics and GPR109A on osteoclast function in cell models[15], we fed BB and HA supplemented diets to male wild type and GPR109A$^{-/-}$ mice for 40 days. Micro-CT determined that bone volume per total tissue volume and trabecular number were significantly higher in GPR109A$^{-/-}$ mice compared with age-matched wild type mice (Fig. 7a, c). Consistent with our previous reports[28,29], wild type mice fed a 5% BB diet had significantly increased bone volume and trabecular number and significantly decreased trabecular separation (Fig. 7b). Although bone volume and trabecular number were still higher and trabecular separation was still lower in BB treated GPR109A$^{-/-}$ mice, statistically there were no differences compared to untreated controls (Fig. 7d). Similarly, all three different doses of

HA significantly increased bone volume and trabecular number and significantly decreased trabecular separation in wild type mice (Fig. 7b). The significant effects of HA on bone volume, trabecular number and trabecular separation were lost in GPR109A$^{-/-}$ mice (Fig. 7d). These data suggest that the effects of BB diet and HA on bone may be partially through GPR109A, with the interpretive caveat that the GPR109A$^{-/-}$ mice already have a positive bone growth phenotype and this may make detection of the effects of phenolics difficult.

We expected that the effects of BB diet and HA on increasing bone mass works through down-regulation of GPR109A in osteoclastic cells to suppress bone resorption. To address this, we isolated proteins from bone marrow cells from wild type and GPR109A$^{-/-}$ mice treated with or without BB diet or HA. Western blots showed that 5% BB and all three doses of HA reduced bone marrow GPR109A protein expression in samples from wild type mice (Fig. 8a). Coincident with this, protein expression of osteoclastic cell differentiation markers Cathepsin K, NFATc1 and MMP9 were all lowered by 5% BB diet and HA treatments in samples from wild type mice (Fig. 8a). However, such inhibitory effects of BB diet and HA on Cathepsin K, NFATc1 and MMP9 protein expression were not found in samples from GPR109A$^{-/-}$ mice (Fig. 8b). Moreover, serum bone remodeling marker measurements indicated that BB diet and HA significantly decreased bone resorption marker CTX-1 levels in wild type mice, but had no effects in GPR109A$^{-/-}$ mice (Fig. 8c). On the other hand, bone formation marker P1NP significantly increased in serum from wild type mice treated with BB diet or HA, but there were no changes in GPR109A$^{-/-}$ mice after BB diet and HA treatment (Fig. 8c). These data again are consistent with our working model that changes in GPR109A

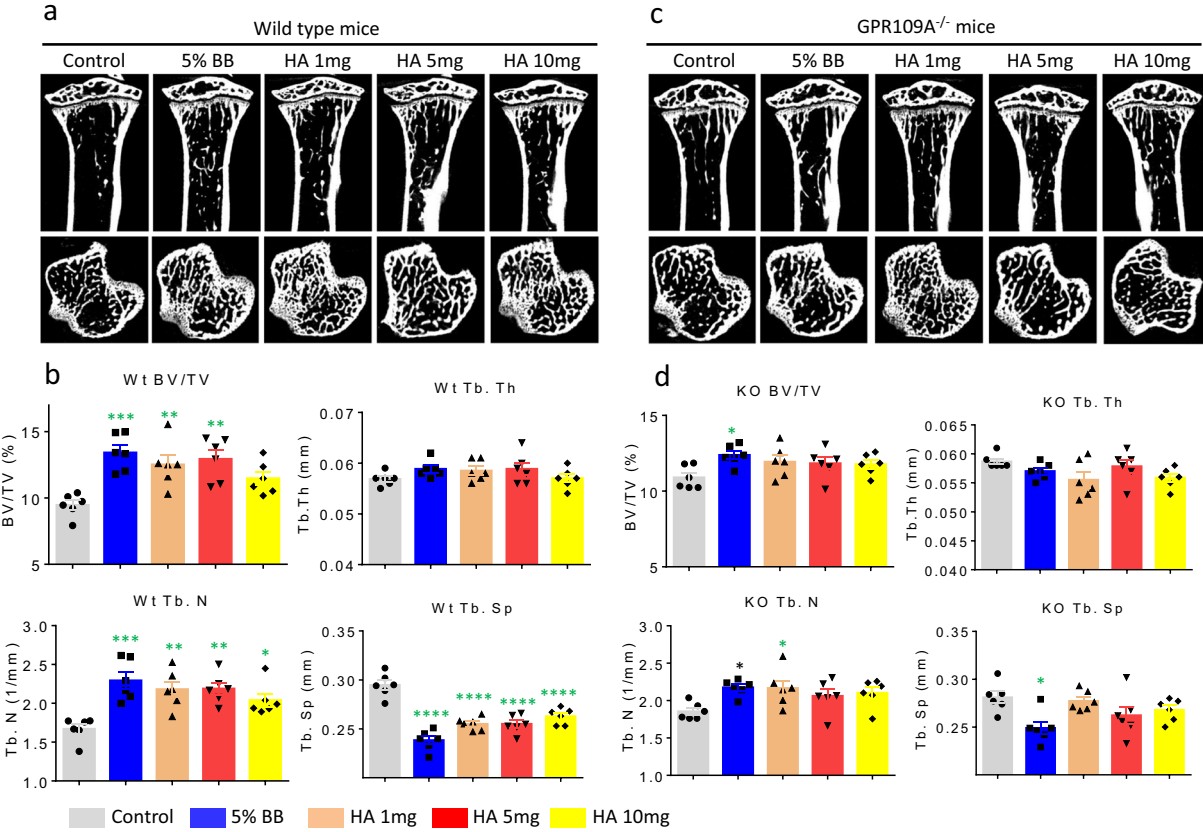

**Fig. 7 BB diet and HA promote bone development in wild type mice but not in GPR109A⁻/⁻ mice.** Four-week-old male wild type and GPR109A⁻/⁻ mice were fed with %5 BB diet and three different doses of HA supplemental diets for 6 weeks. **a** Representative micro-CT images (sagittal view above, axial view below) of the proximal tibial from one sample from each group of wild type mice, white color lines and dots show bone. **b** Micro-CT measured parameters from with or without BB or HA treatments of wild type mouse groups. BV/TV, bone volume / total tissue volume; Tb.Th, trabecular thickness; Tb.N, trabecular number; Tb.Sp, trabecular separation. **c** Representative micro-CT images (sagittal view above, axial view below) of the proximal tibial from one sample from each group of GPR109A⁻/⁻ mice, white color shows bone. **d** Micro-CT measured parameters from with or without BB or HA treatments of GPR109A⁻/⁻ mouse groups. Data are expressed as mean ± SD ($n = 6$ per group); *$p < 0.05$, **$p < 0.01$, ***$p < 0.001$, ****$p < 0.0001$ as determined by one-way ANOVA followed by Student-Newman-Keuls posthoc analysis for multiple pairwise comparisons to control.

action are involved in the bone effects of dietary BB and associated phenolics such as HA.

## Discussion

In the current report, we discovered functional roles of GPR109A in osteoclastogenesis and bone resorption during bone development, with systemic deletion of GPR109A gene leading to increased bone mass and significant changes of microbiota composition in gut in young mice. We presented in vivo and in ex vivo results of hippuric acid (HA) and 3-(3-hydroxyphenyl)-propionic acid (3-3-PPA) effects on inhibiting osteoclastic cell differentiation and bone resorption. The data support the idea that inhibitory effects of HA and 3-3-PPA on osteoclasto-genesis and osteoclast resorption activity involve the cell membrane receptor GPR109A. GPR109A has been shown as an important biomolecular target of niacin, and could therefore be involved in the clinical utility of niacin for the treatment of dyslipidemia and to increase HDL cholesterol[30]. Our data suggested that GPR109A might have additional values and merit further investigations as a target for treatment of metabolic bone disorders. HA and 3-3-PPA are small molecular compounds, their free forms structurally are conjugations of benzoic acid and glycine, and benzene ring conjugated to a propanoic acid. They were found to be the highest levels among the phenolic acids that had at least a 10-fold higher concentration in the serum of rats

fed a BB-containing diet compared to those fed a control diet[14]. The anti-osteoclastogenic bone resorptive properties of HA and 3-3-PPA are recommended for further investigations as alternative compounds for treatment and prevention of osteoclastic bone resorptive disorders such as osteoporosis and rheumatoid arthritis.

G protein coupled receptors (GPCRs) are cell-surface molecules involved in signal transmission and cell differentiation and have emerged as crucial players in child development, growth, and maturation[31,32]. They are the largest and most versatile receptor family and some of them are targets for many approved drugs and those under development[33]. Notably, there are still more than 100 orphan GPCRs for which endogenous ligands are unknown. Several subtypes of GPCRs have been shown specifically in the involvement of skeletal growth[34], and it has been reported that the free fatty acid receptor G protein-coupled receptor 30 (GPR30) protects from bone loss through inhibition of osteoclast differentiation[34]. However, it is not known if GPR109A contributes to skeletal development or plays a role in the osteoclast and osteoblast differentiation pathways. GPR109A was originally identified as a high-affinity receptor for the B-complex vitamin niacin (nicotinic acid)[9]. This provided a molecular mechanism for the niacin-induced correction of dyslipidemia because activation of the receptor in adipocytes inactivates the hormone-sensitive lipase and inhibits lipolysis. We have confirmed that GPR109A is expressed in a wide variety of tissues/

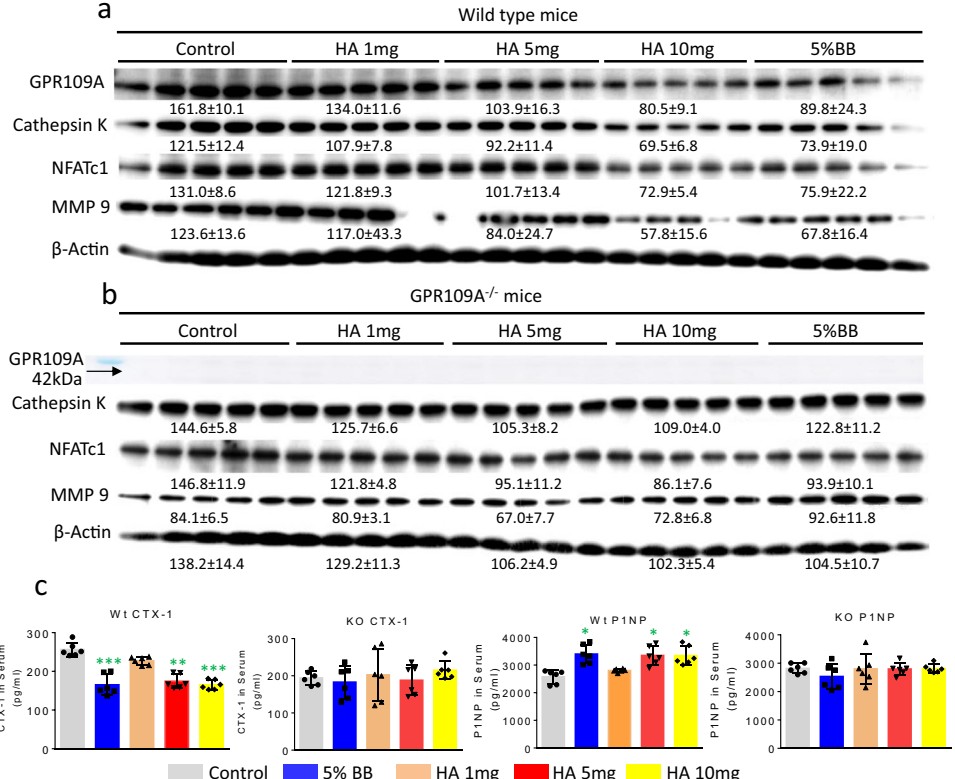

**Fig. 8 HA and 5% BB diet inhibit GPR109A expression and bone resorptive signals in ten weeks old wild type but not in GPR109A$^{-/-}$ male mice.** Proteins were isolated from aspirated femur bone marrow cells, 5 per group from 4-weeks-old wild type or GPR109A$^{-/-}$ mice with or without BB or HA treatments for six weeks. **a** Western blot shows HA and BB diet inhibited GPR109A protein expression, and osteoclast bone resorptive markers Cathepsin K, NFATc1 and MMP9 protein expression in samples from wild type mice. **b** Western blot shows HA and BB diet did not change osteoclast bone resorptive markers Cathepsin K, NFATc1 and MMP9 protein expression in samples from GPR109A$^{-/-}$ mice compared with samples from untreated GPR109A$^{-/-}$ mice. **c** CTX-1 and P1NP levels in serum from wild type (Wt) and GPR109-/- (KO) mice. Data are expressed as mean ± SD ($n = 6$ per group); *$p < 0.05$, **$p < 0.01$, ***$p < 0.001$, as determined by one-way ANOVA followed by Student-Newman-Keuls posthoc analysis for multiple pairwise comparisons to control.

cells including adipose tissue, skin, hepatocytes, retinal cells, and bones; it is also highly expressed in immune cells such as macrophages and dendritic cells[35]. Upon ligand binding, GPR109A couples through the Gi pathway in some tissue cells, resulting in decreased levels of cAMP inside the cells[24,36]. The receptor may function as a suppressor of inflammation and carcinogenesis in the colon[37]. Deletion of the receptor in mice may effect on the progression of colonic inflammation and colon cancer in multiple experimental model systems[12]. Conflicting results on GPR109A expression and function in hepatocytes were also reported recently[13].

Notably, we did not find significant differences in body weights and tibia length (Table 2) between systemic GPR109A gene knockout and wild type rapidly growing young mice. Despite this, we found that systemic GPR109A gene deletion leads to significantly decreased osteoclast formation and increases bone mass. The mechanisms underlying this inhibitory effect of GPR109A on osteoclastogenesis may involve Wnt/β-catenin signaling through inhibition of cAMP entering osteoclast precursors to promote osteoclast differentiation. In fact, systemic GPR109A gene deletion did not completely inhibit osteoclast formation, as GPR109A gene deletion mice were not osteopetrotic, indicating GPR109A is not essential for osteoclastogenesis. Other indirect pathways might be affecting on osteoclast differentiation after GPR109A gene deletion. These remaining questions will be examined in our future studies using bone cell type-specific GPR109A deletion mouse models.

It has been reported that membranous GPR109A is also expressed on mucosal immune cells, particularly in gut epithelial cells and dendritic cells. Activation of the receptor in dendritic cells promotes the ability of these cells to convert naïve T cells into immune suppressive Tregs and potentiates the production of the anti-inflammatory cytokine IL-10; GPR109A-null mice have reduced number of Tregs in the colon, reduced levels of IL-10, and increased levels of the pro-inflammatory cytokine IL-17[35]. Herein, osteoclast numbers were found to be significantly decreased both in vivo and in ex vivo bone marrow cell cultures, it was not clear if this was the case in reducing transition from pre-osteoclastic macrophages to mature osteoclasts after deletion of GPR109A. Recent studies show that NLRP3-mediated inflammasome plays an important role in IL-18 secretion and regulation of microbiota through GPR109A[38]. The latter triggered us to compare gut microbiota of GPR109A gene deletion and wild type mice. The transformations of microbiota composition were significant between GPR109A gene knockout and wild type mice. In both sexes, the β-diversity and microbiota in family level were significantly different between GPR109 knockout and wild type mice. We have not found evidence if the different composition of microbiota between GPR109A and wild type are associated with changes of bone mass and osteoclastic bone resorption. However, it is known that gut microbiota produce several kinds of short chain fatty acids, for example butyrate. It is possible that some short chain fatty acids may be particularly produced after GPR109A deletion in the gut, and they may reach

into target tissue, and have effects on osteoclastogenesis[39]. Such a hypothesis will need to be further evaluated, by (for example) generating a gut epithelial cell-specific GPR109A gene deletion mouse model, and to research specific links among gut microbiota changes, metabolite production, and bone osteoclast resorption.

Research into the correlation between fruit and vegetable intake and bone mineral density has suggested a role for polyphenolic compounds found in fruits and vegetables in promoting bone health[40]. Based on our research, some of those compounds may have potential for long-term value as alternatives to chemically-synthesized medicines for boosting peak bone formation. However, previous reports on phenolic acids' bone protective, antioxidant and anti-inflammatory effects have been highly variable: different phenolic acids may have diverse or opposite effects on osteoblast differentiation or on decreasing the formation of osteoclast-like cells[41]. None of those phenolic acids have been described their mechanisms on either promoting osteoblast differentiation or suppressing osteoclastogenesis. We have previously provided evidence that HA and 3-3-PPA were capable to drive stem cell differentiation potential to specific cell lineage, and might have a profound impact on bone health promotion[19,20]. Circulating HA and 3-3-PPA are thought to be originated from gut microflora-derived metabolites of polyphenols following consumption of BB and other fruits, vegetables, and coffee[42]. They are structurally similar to nicotinic acid or niacin, and bind to GPR109A[20]. In our current study, we found both HA and 3-3-PPA significantly inhibited GPR109A expression in bone marrow cell-derived osteoclast progenitors. Taken together with our previous reports[19,20], we believe that both HA and 3-3-PPA have anabolic effects on bone, and GPR109A may have tissue or cell type-specific roles. We chose a young rapidly growing mouse model to study HA and BB diet in vivo, aiming to see whether HA and BB diet promotes bone growth during early life and whether the effects are mediated through GPR109A. Our data highlighted the inhibitory effects of BB-derived compounds on bone resorption, and how these effects were not seen following whole-body knockout of GPR109A.

In conclusion, we have characterized the effects of systemic deletion of GPR109A on suppression of osteoclastogenesis and bone resorption. We presented that deletion of GPR109A gene leads to increased bone mass and significant changes of microbiota composition in gut in young mice. We suggest that HA and 3-3-PPA effects on inhibiting osteoclastic cell differentiation and bone resorption involves (through as-yet unclear mechanisms) the membranous GPR109A in osteoclast precursors. Studying the intersections of diet-derived compounds such as HA and 3-3-PPA with GPR109A provides new insights about nutritional and other strategies that promote bone health and mitigate or prevent degenerative bone disorders.

## Methods

**Materials and reagents**. Commercially available materials and reagents are listed in Supplementary Table 1, and primer sequences used for real-time PCR are listed in Supplementary Table 2.

**Mouse experiments and in ex vivo cell culture**. GPR109A$^{-/-}$ mice were from Dr. Muthusamy Thangaraju (Department of Biochemistry and Molecular Biology, Georgia Regents University, US)[37], and this systemic GPR109A$^{-/-}$ mouse model was originally made by Dr. Klaus Pfeffer's research group at the Institute of Medical Microbiology and Hospital Hygiene Heinrich-Heine-University Düsseldorf, Germany[1]. We have inbred GPR109A gene knockouts and C57BL/6 J wild type mice to generate male and female GPR109A gene deletion and wild type mice for the current studies. Experiments involved 4-week-old and 6-month-old littermates of male and female GPR109A$^{-/-}$ and their corresponding wild type male and female mice (6 per group) for bone phenotyping studies. Thirty male GPR109$^{-/-}$ mice and thirty male wild type mice were generated for blueberry (BB)

diet and HA supplemental diet feeding studies (6 per treatment group). Mice were weighed, randomized by their weights, and housed 6 per cage in mouse small shoebox cages. Control mice received AIN-93G diet formulated with casein as the sole protein source throughout the experiment. The other eight groups of mice received either 5% BB diet or HA (from Alfa Aesar, USA cas#621-54-5) supplemented 1 mg/kg/day (8.4 mg/kg in diet), 5 mg/kg/day (42 mg/kg in diet) and 10 mg/kg/day (84 mg/kg in diet) daily for 40 days, designated as 5% BB, HA 1, 5 and 10 mg groups, respectively. Mice were housed in an Association for Assessment and Accreditation of Laboratory Animal Care-approved animal facility in the Arkansas Children's Nutrition Center Animal Studies Core at the Arkansas Children's Research Institute, with constant humidity and lights on from 06:00–18:00 hr at 22 °C. All animal procedures were approved by the Institutional Animal Care and Use Committee at University of Arkansas for Medical Sciences (AUP#3595 UAMS, Little Rock, AR). At the end of the studies, mice were anesthetized by injection with 100 mg Nembutal/kg body weight (Avent Laboratories). Blood was collected via cardiac puncture, which was followed by decapitation, femur, tibia and vertebrae bones were collected and stored in −80 °C. Non-adherent bone marrow cells were cultured in 96-well plates (2 × 104 cells/well) in the presence or absence of 50 ng/ml of RANKL, in α-MEM supplemented with 10% fetal bovine serum (FBS) (Hyclone, Logan, UT), penicillin (100 Units/ml), streptomycin (100 μg/ml), and glutamine (4 mM).

**Bone analysis, histology, and microbial community profiling using 16 S rRNA amplicon sequencing and bioinformatics analysis, and other standard methods**. Sample preparation and detailed methods for bone remodeling marker measurements, real-time PCR, and Western blotting are presented in the Supplementary Methods. 16 S rRNA sequencing data were deposited into a public repository of NCBI's Sequence Read Archive database with accession number of GSE161772.

**Statistics and reproducibility**. Statistical power was computed based on a two-factor ANOVA with 6 mice per group. Statistical analysis was performed with GraphPad Prism 8.0 (GraphPad Software, Inc., San Diego, Ca, USA). Numerical variables were expressed as means ± STDEV (Standard Deviation). Comparisons between groups were performed with the nonparametric Kruskal-Wallis test followed by a Dunnett's test comparing each dose to the control group. The nonparametric Wilcoxon rank-sum test was used to compare controls to individual treatment. The rationale for using 6 mice per group is described in the Supplementary Methods. Cell culture experiments were conducted at least three independent times, and representative images are displayed. Dose-response was assessed using Cruick's non-parametric test for trend. The critical $p$-value for statistical significance was[43–54] $p = 0.05$.

**Reporting summary**. Further information on research design is available in the Nature Research Reporting Summary Linked to this article.

## Data availability

16 S rRNA sequencing data were deposited into the NCBI's Sequence Read Archive database with accession number of GSE161772. Source data underlying plots shown in figures are available in Supplementary Data 1. Full blots are shown in Supplementary Information. Additional data related to the paper are available from the corresponding author on reasonable request.

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

## Acknowledgements

This study was supported by USDA-ARS Project 6026-51000-010-05S to Arkansas Children's Nutrition Center and to Dr. Jin-Ran Chen. NIH/NCI 1R01 CA236814-01A1and Department of Defense (DoD) CA180190 grants partially supported the study. We would like to thank the following people for their technical assistance: Matt Ferguson, James Sikes, Hoy Pittman and Bobby Fay.

## Author contributions

J.R.C. is a senior author designed and performed the study and wrote the paper; H.Z. and O.P.L. performed cell, biochemical and molecular experiments; U.W. and K.S. contributed in the analysis of microbiome data. S.V.C. contributed on bioinformatics data analysis. C.L. and D.G. performed and contributed in ex vivo and in vitro osteoclast analysis. F.Z. contributed in the analysis of in vitro experiments.

## Competing interests

The authors declare no competing interests.
