## [Peer Review File · Communications Biology]

Reviewers' Comments:

Reviewer #1:

Remarks to the Author:

This study investigates the role of GPR109A on osteoclasts in vitro and in vivo. This manuscript is well written and logically structured. They use GPR109A knock-out mice to show that they have low numbers of osteoclasts and high bone mass. This defect seems to be cell intrinsic, as also ex vivo differentiated osteoclasts from these mice differentiate less well. They further show that the microbiome is altered by GPR109A knock-out. Finally, they identify metabolites that are associated with GPR109A expression and find that treatment with hippuric acid inhibits osteoclast formation and increases bone mass in wildtype mice. The methods of this study seem well done, although in a few cases, quantification is missing. Overall, an interesting study, although the mechanistic links are not that well defined.

I have the following specific questions:

1. The authors only analyzed 4-week-old mice. These mice are still growing. Mice of at least 12 weeks of age should be bone phenotyped to receive better information on the role of GPR109A on bone remodeling. In this study, they only consider bone growth and modeling.
2. Along those lines, more detailed histological studies should be performed to address the effect of GPR109A on bone. Three samples per group is not much. I would suggest to measure all available samples. The osteoclast data should be quantified properly. The same slides could be used to quantify the number of osteoblasts to obtain a better insight into bone remodeling. I also recommend measuring P1NP in the serum again to obtain some insights into what is happening in parallel to the osteoblasts.
3. Biomechanical testing could be performed if bones are available to unequivocally determine bone strength.
4. I would move the pQCT data from Fig. 3 to Fig. 2. In general, it is unclear, why pQCT was done considering μ CT is much more accurate. The data could even be shifted into a supplement.
5. On which day osteoclasts were the RNA expression measurements done? Please specify.
6. It is unclear what the gut data have to do with the whole story. I know it is trendy, but I wonder if this finding is necessary in this context. Do the authors think there is a link between the microbiome changes and the bone changes? This could be linked better.
7. Again in Fig. 7, histology should be performed to quantify what happens to the bone cells after BB or HA treatment.
8. Is the WB shown in Fig. 1A one WB? It looks like it was cut quite substantially. This applies to most Western blots shown here. I would recommend to add the original files in a supplement. In all cases, the WB should be quantified as well.

Reviewer #2:

Remarks to the Author:

In this manuscript, Chen and colleagues have tried to characterize the phenotype of GPR109A knockout mice and tried to delineate the role of this receptor in HA and 3-3-PPA actions in terms of promoting bone formation. Overall, it appears that the 4 week old GPR109A knockout mice have higher bone mass compared to wild type mice both in tibia and spine. Authors also show that the effect is likely at the level of osteoclast numbers being regulated by GPR109A. Authors further show that HA (hippuric acid) inhibits bone resorption in WT mice and thus increased bone mass. This effect was suppressed in the KO suggesting that HA acts at least partially via GPR109A. Overall, studies are interesting but there are some major concerns as follows:

General major concerns:

This paper is mostly characterizing the phenotype of GPR109A and does not go further in terms of mechanisms by which GPR109A regulates bone. This is a major weakness.

All studies are done in 4 week old mice and thus the effects in adult mice is still unclear.

The effect of hippuric acid and 33-PPA while interesting, the authors don't quite provide mechanistic information in terms of how GPR109A is involved in their actions. Clearly, they regulate GPR109A expression but in the absence of the receptor, 5% BB diet appears to still have effects in the KO and there are some partial effects with HA. As the authors state in the conclusion, it appears that HA and 33PPA somehow involve GPR109A but how exactly is not clear and is needed for making this manuscript mechanistic.

Specific major concerns:

Mice: It is not clear if the mice are littermate (WT and KO). If not, then the differences in microbiota could be due to different cages, rooms etc. Authors should specify these types of information in the methods. Also please make sure to mention if the KO mice have been backcrossed sufficient times to C57BL6 background.

Since these are growing mice, authors should also provide data on bone length.

While it is understandable that the authors focused on osteoclasts, they should provide markers of osteoblasts to provide a complete picture of the knockout mice.

In Fig 8B, as a control, WB of GPR109A should be shown for the KO mice (to show the absence).

Reviewer #3:

Remarks to the Author:

In this article Chen et al show that *Gpr109a*^{-/-} mice have higher bone density than WT mice. They link this finding to decreased number of osteoclast and levels of TNF- α , TRAP, cathepsin K and extracellular cAMP in the former. Additionally they show that diet derived phenolic acids, HA and 3-3-PPA inhibited the expression of *Gpr109a* and bone resorption in a *Gpr109a*-dependent manner. Overall this is an interesting study with several concerns.

1) Fig 1A. GPR109a is expressed at highest levels in adipocytes and innate immune cells (pls see various earlier publications from Dr. Offermanns and others) . In all the other tissue, it is so low that in *Gpr109a*-promoter driven RFP transgenic mice, Dr. Offermanns's group could not directly visualize RFP. Therefore how come all the other tissue in Fig 1A have similar levels of *Gpr109a* expression as adipose tissue and RAW cells. *Gpr109a* is single exon gene. Almost all the RNA – preps are contaminated with DNA. In this scenario, RT-PCR primers will not discriminate between DNA and RNA, and therefore will not accurately depict the expression of *Gpr109a*. Although authors write that RNA was depleted of DNA. To ascertain whether DNA was completely depleted they should do compare the RT-PCR with equivalent amount of RNA and cDNA. In addition, show the expression by quantitative realtime-PCR.

2) Fig 3D. Signaling through *Gpr109a* inhibits intracellular cAMP not extracellular cAMP as shown here. In my knowledge there is neither a report nor a rationale in the scientific literature to explain how Gi-coupled receptors influence extracellular cAMP.

3) Fig 4A shows development of lower numbers of osteoclasts in total non-adherent cells from BM of *Gpr109a*^{-/-} mice. This non-adherent pool of cells was used in Fig 4B-D to show lower bone resorption by *Gpr109a*^{-/-} osteoclasts. Since, cells from *Gpr109a*^{-/-} mice had lower number of osteoclasts, it is completely unclear whether lower bone resorption by *Gpr109a*^{-/-} cells is due to lower number of osteoclasts or their impaired activity. Therefore, authors cannot interpret lower bone resorption activity by *Gpr109a*^{-/-} osteoclasts in absence of any data.

4) There is no experiment to show how gut microbiota is altered or altered gut microbiota affects osteoclast or higher bone density in *Gpr109a*^{-/-} mice.

5) Authors have *Gpr109a*^{-/-} mice. They must authenticate the *Gpr109a* antibody used for various western blot experiments such as Fig 6 and 8.

6) Mechanistically, what do the authors want to convey; Whether HA and 33-PPA binds to *Gpr109a*

to mediate all the effects shown in manuscript or they suppress the Gpr109a expression to mediate their effects. A clear mechanistic data to discriminate between these possibilities are required.

Reviewer 1

Reviewer: This study investigates the role of GPR109A on osteoclasts in vitro and in vivo. This manuscript is well written and logically structured. They use GPR109A knock-out mice to show that they have low numbers of osteoclasts and high bone mass. This defect seems to be cell intrinsic, as also ex vivo differentiated osteoclasts from these mice differentiate less well. They further show that the microbiome is altered by GPR109A knock-out. Finally, they identify metabolites that are associated with GPR109A expression and find that treatment with hippuric acid inhibits osteoclast formation and increases bone mass in wildtype mice. The methods of this study seem well done, although in a few cases, quantification is missing. Overall, an interesting study, although the mechanistic links are not that well defined.

Author's response: We thank this reviewer very much for his/her encouraging comments, we have quantified necessary data, and have provided additional data to explain mechanisms in our revised manuscript.

Reviewer: The authors only analyzed 4-week-old mice. These mice are still growing. Mice of at least 12 weeks of age should be bone phenotyped to receive better information on the role of GPR109A on bone remodeling. In this study, they only consider bone growth and modeling.

Author's response: We have additionally analyzed 6 months old male and female wild type and GPR109A knockout mice femur using three-point-bending method after micro-CT scans (20 slices) to evaluate bone strength. We have presented new data in a new Figure 3 of revised manuscript.

Reviewer: Along those lines, more detailed histological studies should be performed to address the effect of GPR109A on bone. Three samples per group is not much. I would suggest to measure all available samples. The osteoclast data should be quantified properly. The same slides could be used to quantify the number of osteoblasts to obtain a better insight into bone remodeling. I also recommend measuring P1NP in the serum again to obtain some insights into what is happening in parallel to the osteoblasts.

Author's response: Three samples per group were presented as representative images, all other data were at least n=6 per group. We have quantified osteoclast and osteoblast numbers and P1NP levels in the serum and real-time PCR for other bone remodeling markers. New data showed GPR109A systemic gene deletion mice not only have decreased osteoclastic bone resorption, but also have relatively higher osteoblastic bone formation.

Reviewer: Biomechanical testing could be performed if bones are available to unequivocally determine bone strength.

Author's response: We have performed three-point-bending test on 4 weeks old and 6 months old mice, presented as new Figure 3 in the revised manuscript.

Reviewer: **I would move the pQCT data from Fig. 3 to Fig. 2. In general, it is unclear, why pQCT was done considering μ CT is much more accurate. The data could even be shifted into a supplement.**

Author's response: We understand the reviewer's suggestions, and agree with the reviewer's point that μ CT is much more accurate, pQCT was only considered as confirmatory method. However, we were surprised that even though pQCT is less sensitive it still showed clear differences on bone density between wild type and knock out mice, therefore, we kept these data in the figure.

Reviewer: **On which day osteoclasts were the RNA expression measurements done? Please specify.**

Author's response: RNA expression measurements for osteoclast were performed after 48 hours of culture, we have specified.

Reviewer: **It is unclear what the gut data have to do with the whole story. I know it is trendy, but I wonder if this finding is necessary in this context. Do the authors think there is a link between the microbiome changes and the bone changes? This could be linked better.**

Author's response: Until now, the functions of GPR109A were mostly reported in immune and GI systems and as a receptor for Niacin, we originally thought that GPR109A gene deletion might change gut microbiome composition therefore linking to osteoclastic macrophage function. We will need to perform many more experiments to determine the link between the microbiome changes and the bone changes. Nonetheless, we think these microbiome data are very unique and original; therefore, they were presented as supplemental figure in our current manuscript for our future reference.

Reviewer: **Again in Fig. 7, histology should be performed to quantify what happens to the bone cells after BB or HA treatment.**

Author's response: We agree with the reviewer's suggestion. Using histomorphometric method, we previously well-quantified bone cells after BB and HA treatment in wild type mice, and due to COVID-19 and significant labor disruption in my lab, we will present such data in our next GPR109A osteoclastic cell specific knockout mouse model. Instead, we have measured CTX-1 and PINP levels in serum, this was a lot of work, and these data were presented in the revision of Figure 8 C.

Reviewer: 2

Reviewer: In this manuscript, Chen and colleagues have tried to characterize the phenotype of GPR109A knockout mice and tried to delineate the role of this receptor in HA and 3-3-PPA actions in terms of promoting bone formation. Overall, it appears that the 4 week old GPR109A knockout mice have higher bone mass compared to wild type mice both in tibia and spine. Authors also show that the effect is likely at the level of osteoclast numbers being regulated by GPR109A. Authors further show that HA (hippuric acid) inhibits bone resorption in WT mice and thus increased bone mass. This effect was suppressed in the KO suggesting that HA acts at least partially via GPR109A. Overall, studies are interesting but there are some major concerns as follows.

Author's response: We thank this reviewer for spending his/her important time reviewing our work, and giving encouraging comments. We have addressed the reviewer's major concerns in the revision.

Reviewer: This paper is mostly characterizing the phenotype of GPR109A and does not go further in terms of mechanisms by which GPR109A regulates bone. This is a major weakness.

Author's response: We thank this reviewer for pointing out the weakness of our current report. We have provided mechanistic evidence underlying inhibitory effect of GPR109A on osteoclastogenesis that might involve Wnt/ β -catenin signaling in pre-osteoclasts through inhibition of secondary messenger of cAMP in osteoclast precursors.

Reviewer: All studies are done in 4 week old mice and thus the effects in adult mice is still unclear.

Author's response: We have additionally analyzed 6 months old male and female wild type and GPR109A knockout mice femur using three-point-bending method after micro-CT scans (20 slices) to evaluate bone strength. We have presented new data in a new Figure 3 of revised manuscript.

Reviewer: The effect of hippuric acid and 33-PPA while interesting, the authors don't quite provide mechanistic information in terms of how GPR109A is involved in their actions. Clearly, they regulate GPR109A expression but in the absence of the receptor, 5% BB diet appears to still have effects in the KO and there are some partial effects with HA. As the authors state in the conclusion, it appears that HA and 33PPA somehow involve GPR109A but how exactly is not clear and is needed for making this manuscript mechanistic.

Author's response: We have previously presented that HA and 3-3-PPA binds to GPR109A but not GPR109B with relative high affinity (JBMR Plus, 2019 Aug

23;3(9):e10201). The mechanisms of HA and 3-3-PPA on inhibition of osteoclastogenesis were most likely through inhibition of GPR109A expression leading to down regulation of Wnt/beta-catenin signaling in pre-osteoclasts (J Cell Physiol. 2020 Jan;235(1):599-610). We have discussed that the effects of 5% BB diet in KO mice may be due to biologically active compounds contained in 5% BB diet other than HA and 3-3-PPA.

Reviewer: **Specific major concerns:**

Mice: It is not clear if the mice are littermate (WT and KO). If not, then the differences in microbiota could be due to different cages, rooms etc. Authors should specific these types of information in the methods. Also please make sure to mention if the KO mice have been backcrossed sufficient times to C57BL6 background.

Author's response: The wild type and KO mice used in the current study were littermates. Although they were in different cages, conditions of cages, rooms and others were exactly matched and same. We have described these conditions of animal housing in our revised manuscript.

Reviewer: **Since these are growing mice, authors should also provide data on bone length.**

Author's response: We have provided in the revision (Figure 4).

Reviewer: **While it is understandable that the authors focused on osteoclasts, they should provide markers of osteoblasts to provide a complete picture of the knockout mice.**

Author's response: We have provided osteoblastic markers (P1NP) in the revision (Figure 4 and Figure 9).

Reviewer: **In Fig 8B, as a control, WB of GPR109A should be shown for the KO mice (to show the absence).**

Author's response: We have provided information of GPR109A in Fig 8A in the revision.

Reviewer: 3

Reviewer: In this article Chen et al show that Gpr109a^{-/-} mice have higher bone density than WT mice. They link this finding to decreased number of osteoclast and levels of TNF- α , TRAP, cathepsin K and extracellular cAMP in the former. Additionally they show that diet derived phenolic acids, HA and 3-3-PPA inhibited the expression of Gpr109a and bone resorption in a Gpr109a-dependent manner. Overall this is an interesting study with several concerns.

Author's response: We thank this reviewer for spending his/her time reviewing our work, and giving encouraging comments. We have addressed the reviewer's concerns in the revision.

Reviewer: Fig 1A. GPR109a is expressed at highest levels in adipocytes and innate immune cells (pls see various earlier publications from Dr. Offermanns and others). In all the other tissue, it is so low that in Gpr109a-promoter driven RFP transgenic mice, Dr. Offermann's group could not directly visualize RFP. Therefore how come all the other tissue in Fig 1A have similar levels of Gpr109a expression as adipose tissue and RAW cells. Gpr109a is single exon gene. Almost all the RNA – preps are contaminated with DNA. In this scenario, RT-PCR primers will not discriminate between DNA and RNA, and therefore will not accurately depict the expression of Gpr109a. Although authors write that RNA was depleted of DNA. To ascertain whether DNA was completely depleted they should do compare the RT-PCR with equivalent amount of RNA and cDNA. In addition, show the expression by quantitative realtime-PCR.

Author's response: GPR109A^{-/-} mice were originally from Dr. Offermanns' group, we not only checked their every publications, but also contacted them about using this mouse model for studying skeletal tissue. As the reviewer suggested, we have additionally performed real-time PCR and quantified GPR109A expression in different tissues. It is exactly as expected, GPR109A is expressed at the highest levels in adipocytes and innate immune cells relative to other tissues and cell types (normalized by GAPDH control). We have attached data in Fig 1A in the revision.

Reviewer: Fig 3D. Signaling through Gpr109a inhibits intracellular cAMP not extracellular cAMP as shown here. In my knowledge there is neither a report nor a rationale in the scientific literature to explain how Gi-coupled receptors influence extracellular cAMP.

Author's response: We proposed that inhibited intracellular cAMP through GPR109A is one of the mechanisms leading to decreased osteoclastogenesis, it was not extracellular cAMP, we corrected in the revision.

Reviewer: Fig 4A shows development of lower numbers of osteoclasts in total non-adherent cells from BM of Gpr109a^{-/-} mice. This non-adherent pool of cells was

used in Fig 4B-D to show lower bone resorption by Gpr109a^{-/-} osteoclasts. Since, cells from Gpr109a^{-/-} mice had lower number of osteoclasts, it completely unclear whether lower bone resorption by Gpr109a^{-/-} cells is due to lower number of osteoclasts or their impaired activity. Therefore, authors cannot interpret lower bone resorption activity by Gpr109a^{-/-} osteoclasts in absence of any data.

Author's response: We agree, but we have also measured bone resorption markers, and osteoclast numbers *in vivo* (Figure 4)

Reviewer: There is no experiment to show how gut microbiota is altered or altered gut microbiota affects osteoclast or higher bone density in Gpr109a^{-/-} mice.

Author's response: Reviewer was right, we presented original and unique data, please allow us to address links between changes of gut microbiota and osteoclast or higher bone density in Gpr109a^{-/-} mice in our next report using osteoclast specific GPR109A knockout mouse model.

Reviewer: Authors have Gpr109a^{-/-} mice. They must authenticate the Gpr109a antibody used for various western blot experiment such as Fig 6 and 8.

Author's response: We thank the reviewer the suggestion, we have performed in the revision.

Reviewer: Mechanistically, what do the authors want to convey; Whether HA and 3-3-PPA binds to Gpr109a to mediate all the effects shown in manuscript or they suppress the Gpr109a expression to mediate their effects. A clear mechanistic data to discriminate between these possibilities are required.

Author's response: We have previously presented that HA and 3-3-PPA binds to GPR109A but not GPR109B with relative high affinity (JBMR Plus, 2019 Aug 23;3(9):e10201). The mechanisms of HA and 3-3-PPA on inhibition of osteoclastogenesis were most likely through inhibition of GPR109A expression leading to down regulation of Wnt/beta-catenin signaling in pre-osteoclasts (J Cell Physiol. 2020 Jan;235(1):599-610).

We thank all the reviewers for their time and thoughtful comments. We believe that the resultant revisions have significantly strengthened the manuscript.

Yours Sincerely

Jin-Ran Chen, M.D., Ph.D.
Associate Professor

Communications Biology MS# COMMSBIO-20-0364-T

College of Medicine
Department of Pediatrics
University of Arkansas for Medical Sciences
Arkansas Children's Nutrition Center
Arkansas Children's Research Institute

Reviewers' Comments:

Reviewer #1:

Remarks to the Author:

The authors have addressed my concerns. I only have minor comments for improvement:

1. Add age in Figure 2 and Table 1; similarly for Fig. 4E and F
2. Comment on bone resorption: the CTX data show bone resorption, as it measures collagen breakdown. In vitro however, it looks like no osteoclasts differentiate at all - thus, it is not surprising to not find resorbed areas on the bone chips. I think this should be considered then describing the data.

Reviewer #2:

Remarks to the Author:

This is a revised manuscript. Authors have addressed most of my previous concerns.

Addition to osteoblast data is interesting but the authors don't quite provide enough discussion of whether the changes in osteoblast numbers and markers are direct or indirect effect of the receptor knockout. In addition, the authors talk about intracellular cAMP data in the rebuttal but that data is not shown.

Reviewer #3:

Remarks to the Author:

Authors have addressed all the concerns. The revised manuscript is suitable for publication in Communications Biology